# Emerging Trends in AI and Radiomics for Bladder, Kidney, and Prostate Cancer: A Critical Review

**DOI:** 10.3390/cancers16040810

**Published:** 2024-02-16

**Authors:** Georgios Feretzakis, Patrick Juliebø-Jones, Arman Tsaturyan, Tarik Emre Sener, Vassilios S. Verykios, Dimitrios Karapiperis, Themistoklis Bellos, Stamatios Katsimperis, Panagiotis Angelopoulos, Ioannis Varkarakis, Andreas Skolarikos, Bhaskar Somani, Lazaros Tzelves

**Affiliations:** 1School of Science and Technology, Hellenic Open University, 26335 Patras, Greece; georgios.feretzakis@ac.eap.gr (G.F.); verykios@eap.gr (V.S.V.); 2Department of Urology, Haukeland University Hospital, 5021 Bergen, Norway; jonesurology@gmail.com; 3Department of Clinical, Medicine University of Bergen, 5021 Bergen, Norway; 4European Association of Urology, Young Academic Urologists, Urolithiasis Group, NL-6803 Arnhem, The Netherlands; tsaturyanarman@yahoo.com (A.T.); emre.sener@marmara.edu.tr (T.E.S.); 5Department of Urology, Erebouni Medical Center, Yerevan 0087, Armenia; 6Department of Urology, Marmara University School of Medicine, Istanbul 34854, Turkey; 7School of Science and Technology, International Hellenic University, 57001 Thessaloniki, Greece; dkarapiperis@ihu.edu.gr; 8Second Department of Urology, Sismanoglio Hospital, National and Kapodistrian University of Athens, 15126 Athens, Greece; bellos.themistoklis@gmail.com (T.B.); stamk1992@gmail.com (S.K.); angelopoulospanag@gmail.com (P.A.); medvark3@yahoo.com (I.V.); andskol@yahoo.com (A.S.); 9Department of Urology, University of Southampton, Southampton SO17 1BJ, UK; bhaskarsomani@yahoo.com

**Keywords:** artificial intelligence, radiomics, urological cancers, oncology, bladder cancer, kidney cancer, prostate cancer, diagnostic imaging, personalized medicine

## Abstract

**Simple Summary:**

In an age where technology is deeply intertwined with healthcare, this review focuses on the synergistic role of artificial intelligence (AI) and radiomics in the management of urological cancers, particularly bladder, kidney, and prostate cancers. Our comprehensive review explores how AI’s rapid data-processing capabilities, combined with the intricate image analysis offered by radiomics, are reshaping cancer diagnosis and treatment. We delve into current research findings to illustrate how these innovative technologies are steering oncology toward more accurate, personalized care. This summary is crafted to be accessible, avoiding complex medical jargon and extensive academic references, aiming to highlight the essence and potential impact of these advancements. Our objective is to showcase how AI and radiomics are instrumental in early cancer detection, informed therapeutic decisions, and potentially improved patient outcomes. The research compiled in this paper not only charts a course for the future integration of these technologies in cancer care but also underscores the emerging trend towards patient-centric strategies in the medical community, offering renewed hope and direction in the fight against these cancers.

**Abstract:**

This comprehensive review critically examines the transformative impact of artificial intelligence (AI) and radiomics in the diagnosis, prognosis, and management of bladder, kidney, and prostate cancers. These cutting-edge technologies are revolutionizing the landscape of cancer care, enhancing both precision and personalization in medical treatments. Our review provides an in-depth analysis of the latest advancements in AI and radiomics, with a specific focus on their roles in urological oncology. We discuss how AI and radiomics have notably improved the accuracy of diagnosis and staging in bladder cancer, especially through advanced imaging techniques like multiparametric MRI (mpMRI) and CT scans. These tools are pivotal in assessing muscle invasiveness and pathological grades, critical elements in formulating treatment plans. In the realm of kidney cancer, AI and radiomics aid in distinguishing between renal cell carcinoma (RCC) subtypes and grades. The integration of radiogenomics offers a comprehensive view of disease biology, leading to tailored therapeutic approaches. Prostate cancer diagnosis and management have also seen substantial benefits from these technologies. AI-enhanced MRI has significantly improved tumor detection and localization, thereby aiding in more effective treatment planning. The review also addresses the challenges in integrating AI and radiomics into clinical practice, such as the need for standardization, ensuring data quality, and overcoming the “black box” nature of AI. We emphasize the importance of multicentric collaborations and extensive studies to enhance the applicability and generalizability of these technologies in diverse clinical settings. In conclusion, AI and radiomics represent a major paradigm shift in oncology, offering more precise, personalized, and patient-centric approaches to cancer care. While their potential to improve diagnostic accuracy, patient outcomes, and our understanding of cancer biology is profound, challenges in clinical integration and application persist. We advocate for continued research and development in AI and radiomics, underscoring the need to address existing limitations to fully leverage their capabilities in the field of oncology.

## 1. Introduction

The advent of artificial intelligence (AI) and radiomics in oncology marks a pivotal moment in the evolution of personalized medicine, particularly in the diagnosis and treatment of urological cancers such as bladder, kidney, and prostate cancer. These cutting-edge technologies are reshaping the foundations of cancer care, heralding a paradigm shift towards precision-driven methods. AI’s ability to process vast datasets rapidly is unveiling new prospects in the comprehension of complex cancer dynamics, allowing clinicians to extract meaningful insights for more accurate diagnoses and efficacious treatment strategies [1,2]. 

Despite these advancements, a gap remains in fully understanding the clinical implementation and integration of AI and radiomics in urological cancer care, necessitating a critical review of emerging trends and their tangible impacts on patient outcomes.

Based on a literature search (Figure 1), we can observe distinct trends in the scientific literature related to cancer research on PubMed from 2016 to 2023. Two lines illustrate the volume of published studies: one represents publications associated with “AI and Radiomics”, and the other depicts cancer research without these terms. The data indicate a significant upward trajectory in publications that involve AI and radiomics, reflecting the growing interest and integration of these technologies in cancer research. Conversely, publications without these keywords appear to remain relatively stable over the same period, suggesting a steady but less dramatic growth in general cancer research. This trend underlines the impact of technological advancements in the field of oncology, where AI and radiomics are becoming increasingly central to research, potentially driving innovation in the diagnosis, treatment, and prognosis of cancer. A notable example [1] is the use of a 3D deep radiomics pipeline in analyzing the CT scans of metastatic urothelial cancer patients. This approach differentiated between disease control and progression in response to immunotherapy, demonstrating AI’s potential as a non-invasive biomarker with a predictive accuracy of 82.5%. This underscores AI’s growing significance in personalized cancer care. A recent literature review [2] focuses on the integration of AI-powered radiomics in urologic oncology. It highlights significant advances in diagnostics and prognosis, like improved lesion detection in prostate cancer through machine learning (ML) and using radiomics for differentiating renal masses in kidney cancer. Challenges remain, such as small sample sizes and the need for broader validation.

Radiomics enhances this landscape by offering an intricate analysis of medical images, identifying tumor patterns that escape the naked eye, thereby enriching our methodology in tumor characterization and affording a nuanced perspective of cancer behavior and progression [3]. Schawkat et al. took an in-depth look at novel imaging approaches for evaluating renal masses and renal cell carcinoma [3]. They discussed the updated Bosniak classification and the clear cell carcinoma likelihood score, along with newer modalities like contrast-enhanced ultrasound, dual-energy CT, and molecular imaging [3]. The integration of radiomics and artificial intelligence (AI) techniques was also explored. These contemporary diagnostic tools, combined with established methods, could address current limitations in renal mass characterization [3].

Evrimler et al. in their study assessed the potential of machine learning (ML)-based CT radiomics in predicting histological variants of bladder urothelial carcinoma, which is crucial for management [4]. It involved analyzing texture features from CT scans of 37 tumors, augmented by synthetic data. The study compared 15 ML algorithms, with the best models achieving high predictive accuracy [4].

In the realm of kidney cancer, particularly renal cell carcinoma (RCC), with its inherent heterogeneity, AI and radiomics adeptly address diagnostic and prognostic challenges. ML algorithms, when applied to MRI-derived radiomics features, have shown promise in distinguishing RCC subtypes and grades [5,6]. Cui et al. [5] investigated MR- and CT-based ML models for grading clear cell RCC. They included patients between 2009 and 2018 for model development and validation, with external validation from an independent institution and The Cancer Imaging Archive [5]. The study focused on the reproducibility and accuracy of texture features from MR and CT images, finding that MR- and CT-based models effectively distinguished high- from low-grade ccRCCs. Zhang et al. [6] focused on investigating radiomics features (RFs) related to the progression-free survival of RCC, aiming to develop a nomogram for individualized treatment reference. The research involved analyzing RFs and clinical data from 175 patients, using enhanced CT imaging and the LASSO algorithm for feature selection. The resulting radiomics nomogram integrated RFs and clinical predictors, demonstrating improved predictive accuracy for progression-free survival over clinical variables alone, highlighting its potential for personalized post-operative patient care in RCC [6]. The study’s radiomics nomogram, including age, clinical stage, Karnofsky performance status (KPS) score, and a weighted sum of six RFs, showed good discrimination and calibration. The C-index for the final model was 0.836 in the training set and 0.706 in the validation set, significantly outperforming the clinical-only model. The model’s clinical usefulness was confirmed through decision curve analysis, indicating its potential for guiding post-operative care in RCC patients. Similarly, predictive models derived from imaging data have been constructed to anticipate patient responses to treatments like immunotherapy and targeted agents, thus assisting in crafting personalized therapy plans [7]. 

Prostate cancer (PCa), being the most prevalent cancer in men, has witnessed significant advancements with the application of AI in MRI-based detection and management. Algorithms driven by AI have enhanced the precision in detecting clinically significant PCa, mitigating unnecessary biopsies, and facilitating informed treatment decisions [8,9]. Qiu et al. [8] developed a peritumoral radiomic-based ML model to differentiate between low and high Gleason-grade group lesions. They included 175 patients and used MRI sequences to delineate original and peritumoral regions of interest [8]. The model, which combined peritumoral features, outperformed other models with an area under the curve (AUC) of 0.850 and an average accuracy of 0.950, showing greater efficacy in predicting peripheral zone lesions. In a multi-institutional study [9], radiomic features from 3T mpMRI were analyzed to differentiate PCa detection in the transition and peripheral zones. The study extracted various features from MRI scans. Feature selection identified 10 distinct features for each zone. Zone-specific classifiers significantly improved cancer detection accuracy in the peripheral zone compared to a zone-ignorant classifier with an AUC of 0.61–0.71 across different institutions, highlighting the importance of considering zone differences in PCa diagnosis. Moreover, the amalgamation of AI with radiomics and genomics is carving out new paths in deciphering the molecular intricacies of PCa, potentially leading to more targeted and effective treatments [10,11]. In a study, authors [10] developed a deep learning (DL) algorithm to classify areas of increased uptake in bone scintigraphy scans, commonly used for screening metastatic bone disease (MBD). Trained and validated on scans from three European medical centers, the algorithm demonstrated high sensitivity and specificity (0.82 and 0.80, respectively) on an external test set. It significantly outperformed nuclear medicine physicians in terms of processing time [10]. 

In another study [11], ML models using radiomic features from mpMRI effectively detected and classified PCa in 191 patients who underwent mpMRI and biopsies. In reference to the study [11], Figure 2 exemplifies the meticulous process of image segmentation on mpMRI, which serves as a foundation for the machine learning models employed to detect and categorize prostate cancer. This figure (Figure 2) presents a representative case of a 60-year-old patient with a PSA level of 12.9 ng/mL and a non-suspicious digital rectal exam (DRE). The mpMRI images show the axial T2-weighted and ADC views where manual delineation has been carried out. The blue contour indicates the boundary of the entire prostate gland, capturing the complex anatomy inclusive of both the peripheral and transition zones. The green contour marks the transition zone specifically, while the red contour highlights the index lesion situated in the right peripheral zone at the apex, which was assigned a PI-RADS score of 4 due to its suspicious characteristics.

The precision of these delineations facilitated the extraction of radiomic features, which when combined with clinical assessments such as PI-RADS, PSAD, and DRE, yielded a robust model with high AUC values ranging from 0.844 to 0.889. This integrated approach not only improved the detection of malignant versus benign lesions but also proved effective in differentiating between clinically significant and insignificant prostate cancer. Such detailed segmentation underscores the combined model’s capability to exceed the diagnostic accuracy of the PI-RADS score alone and perform better than the mean ADC values in prognosticating clinically significant prostate cancer, as delineated in [11].

## 2. Materials and Methods

### Criteria for Selecting Articles

For this comprehensive narrative review, a stringent selection process was implemented to identify the most relevant and impactful studies in the field of AI and radiomics as applied to bladder, kidney, and prostate cancers. The primary criteria for article selection included the following:Relevance to AI and Radiomics: We selected studies specifically oriented toward the integration of AI and radiomics in the diagnostics, treatment, and prognosis of bladder, kidney, and prostate cancers. This criterion ensured that the reviewed literature directly contributes to our understanding of the current capabilities and future potential of these technologies in the specific context of urological cancers;Clinical Significance: Given the practical orientation of our review, we prioritized studies that demonstrate substantial clinical relevance. These are studies that offer insights into the real-world applicability of AI and radiomics, such as those leading to improved precision in diagnostic imaging, enhanced predictive accuracy of treatment outcomes, or novel radiomic biomarkers that can inform patient management in bladder, kidney, and prostate cancers;Methodological Rigor: Recognizing the importance of robust research design, we favored studies exemplifying methodological rigor. This included clearly articulated data collection processes, sophisticated and transparent analytical methods, and rigorous validation techniques. This criterion was essential to ensure that our review is grounded in studies that offer reliable and replicable findings, contributing to the solidification of AI and radiomics as cornerstones in oncological research and practice;Recent Publications: To capture the dynamic and rapidly progressing nature of AI and radiomics, we emphasized recent studies published within the last few years. This criterion allowed our review to act as a beacon, highlighting the trajectory of recent innovations and discerning the immediate future of AI and radiomics in the context of urological cancers.

## 3. AI and Radiomics in Bladder Cancer

### 3.1. Recent Advancements

The diagnosis and treatment of bladder cancer have been profoundly transformed by the recent advancements in AI and radiomics. These innovative technologies provide a more nuanced understanding of bladder cancer, offering deeper insights into its characteristics and behavior through enhanced imaging analysis and predictive modeling. AI has revolutionized the diagnostic approach for bladder cancer, particularly when integrated with imaging modalities such as MRI and CT scans [12,13]. A study [12] utilizing a hybrid DL model showed impressive numerical results across three classification tasks: distinguishing normal tissue from bladder cancer, differentiating non-muscle-invasive bladder cancer (NMIBC) from muscle-invasive bladder cancer (MIBC), and identifying post-treatment changes (PTC) from MIBC. For the normal vs. bladder cancer task, the model achieved an accuracy of 86.07%, sensitivity of 96.75%, and specificity of 69.65% [12]. The best performance was observed with the LDA classifier on XceptionNet-based features [12]. In distinguishing NMIBC from MIBC, the accuracy was 79.72%, with a sensitivity of 66.62% and specificity of 87.39% [12]. For the post-treatment changes vs. MIBC task, the model recorded an accuracy of 74.96%, sensitivity of 80.51%, and specificity of 70.22% [12]. 

In an extensive analysis conducted by Sarkar et al., the complex stratification of bladder cancer stages was visually depicted through a series of axial CT scans with intravenous contrast. Figure 3 from Sarkar et al. presents a detailed view of the different stages, ranging from Ta, which represents non-invasive papillary carcinoma, to T4, indicative of advanced cancer with invasion beyond the bladder. The delineation of regions of interest (ROI) on the bladder wall corresponding to each stage emphasizes the potential challenges in visual diagnosis and the suitability of such images for advanced AI-based classification models [12]. As shown in Figure 3 above, the progression from non-muscle-invasive to muscle-invasive bladder cancer, through stages Ta, Tis, T1 (NMIBC) to T2, T3, and T4 (MIBC), underscores the necessity for precise staging, which is crucial for treatment planning and prognostic assessment.

Li et al. in their study [13] aimed to compare the effectiveness of radiomics, single-task DL, and multi-task DL methods in predicting MIBC using T2-weighted imaging (T2WI). It included 121 tumors, with 93 for training and 28 for testing. The AUC values in the training cohort were 0.920 (radiomics), 0.933 (single-task), 0.932 (multi-task), and 0.844, 0.884, and 0.932, respectively, in the test cohort [13]. The study concluded that all models showed good diagnostic performance, with the multi-task model being the most effective and focused on diseased tissue areas [13].

AI’s role in the realm of medical imaging marks a significant leap in oncology, especially in evaluating treatment responses. AI algorithms enable the quantification of radiologic characteristics beyond the capabilities of conventional imaging techniques, equipping clinicians with the ability to extract detailed quantitative data from images. This offers an unprecedented level of precision in diagnosing and monitoring diseases [1]. A notable study in this field utilized a 3D deep radiomics pipeline to analyze chest–abdomen CT scans, aiming to differentiate between disease control and progression in patients undergoing immunotherapy with immune checkpoint inhibitors (ICIs). This study involved monitoring 42 patients with metastatic urothelial cancer post-first-line platinum-based chemotherapy [1]. The deep learning pipeline, incorporating self-learned visual features and a deep self-attention mechanism, demonstrated a predictive accuracy of 82.5%, highlighting AI’s potential as a non-invasive biomarker for predicting disease control in response to ICIs in metastatic urothelial cancer [1].

Radiomics has revolutionized medical imaging, particularly in the analysis and interpretation of disease characteristics. By extracting an array of features from medical images, radiomics allows for an in-depth examination of disease manifestations. Processing these features through AI algorithms reveals complex patterns correlating with disease attributes and patient outcomes, often uncovering associations not apparent through conventional analysis [14]. In the context of bladder cancer (BCa), a significant study focused on constructing a CT-based deep learning radiomics nomogram (DLRN) to predict the pathological grade of BCa pre-operatively. The study, which extracted both handcrafted and DL radiomics features from multi-phase CT images of a large patient cohort, employed 11 machine learning classifiers [14]. The results showed the superiority of the support vector machine (SVM) classifier that combined these features, demonstrating the CT-based DLRN as a powerful diagnostic tool for differentiating between high and low-grade BCa, potentially guiding more precise pre-operative planning and personalized treatment approaches [14]. In another study [15], authors developed a CT-based radiomics model to preliminarily predict bladder-cancer grade. Patients with surgically resected bladder cancer, who had undergone CT urography, were divided into training and validation groups. The model used logistic regression and its performance was evaluated using the ROC curve, AUC, sensitivity, specificity, Positive Predictive Value (PPV), and Negative Predictive Value (NPV) [15]. An AUC of 0.950 in the training group and 0.860 in the validation group was detected. In the validation group, the model achieved 83.8% accuracy, 88.5% sensitivity, 72.7% specificity, 88.5% PPV, and 72.7% NPV, demonstrating its effectiveness. 

### 3.2. Key Findings from Selected Studies

Key studies in this area include the use of CT-based deep-learning radiomics for creating a nomogram for the preoperative prediction of pathological grades in bladder cancer [16]. This research demonstrated the capability of radiomics features from pre-operative CT scans to construct a predictive model with remarkable accuracy, aiding clinicians in treatment planning and decision-making.

Rundo et al. in their study investigated the use of a 3-dimensional deep radiomics pipeline in analyzing CT scans for metastatic urothelial cancer patients [1]. This AI-based approach focused on differentiating disease control from progression in patients undergoing immunotherapy after first-line platinum-based chemotherapy failure [1]. The predictive accuracy of the pipeline was 82.5%, with a sensitivity of 96% and specificity of 60%, while including baseline clinical factors raised the accuracy to 92.5% [1]. This AI approach shows promise as a non-invasive biomarker for predicting responses to immunotherapy in metastatic urothelial cancer [1].

### 3.3. Implications for Diagnosis and Treatment

The advancements in AI and radiomics have substantial implications for bladder cancer diagnosis and treatment. By providing more accurate and comprehensive tumor information, these technologies can significantly refine the precision of diagnoses. This, in turn, facilitates more effective treatment strategies tailored to individual patient needs, particularly crucial in bladder cancer where treatment decisions heavily depend on the tumor’s stage and grade. As these technologies continue to evolve, their incorporation into clinical workflows is expected, heralding a sophisticated and patient-centric approach to bladder cancer management.

The predictive ability of AI and radiomics in forecasting tumor behavior and response to treatment is ushering in a new era of personalized medicine in bladder-cancer care. This promises more efficacious treatment plans with potentially fewer side effects and improved patient outcomes. 

## 4. AI and Radiomics in Kidney Cancer

### 4.1. Technological Innovations

In the realm of kidney-cancer care, especially for RCC, AI and radiomics have ushered in groundbreaking technological advancements. These developments have focused on refining imaging techniques to enhance diagnostic precision, differentiate among RCC subtypes, and predict patient outcomes more accurately. A significant innovation in this domain is the application of ML algorithms, which analyze radiomic features from MRI and CT scans. AI and radiomics play a crucial role in monitoring disease progression and therapy response, ensuring timely adjustments in treatment plans when necessary [17,18]. Roussel et al. [18] aimed to differentiate benign renal tumors from renal cell carcinoma (RCC) using a DL model based on MR imaging. The model applied to MR images of 1162 renal lesions and combined clinical variables with MR images using a ResNet-based ensemble model [18]. It demonstrated significantly higher accuracy (0.70), sensitivity, and specificity compared to both expert interpretations and the best radiomics model. These algorithms are adept at extracting and processing extensive datasets from medical images, surpassing the capabilities of traditional methods [19,20]. A systematic review and meta-analysis evaluated radiomics’ diagnostic accuracy in renal tumor differentiation and treatment response assessment in metastatic RCC (mRCC) [19]. Of 1098 identified studies, 113 met the inclusion criteria. The median radiomics quality score (RQS) of these studies was low but showed improvement over time [19]. The quantitative synthesis of 30 studies demonstrated significant odds ratios for differentiating various benign renal tumors from RCC [19]. However, disparate study designs regarding mRCC treatment response precluded a meaningful meta-analysis. The findings highlight radiomics’ potential in renal tumor characterization, emphasizing the need for shared data and collaborative research to enhance study reproducibility and reliability [19].

Radiogenomics marks a pivotal shift in personalized medicine, merging the visual detail of radiomic data with genomic insights to form a comprehensive view of cancerous tumors. This fusion enables clinicians to examine both the phenotype—the observable characteristics of the tumor—and the genotype—the genetic profile underpinning tumor behavior. The application of radiogenomics is transformative in oncology, where treatments and prognoses are increasingly tailored to each patient’s unique tumor characteristics. A notable study in this area involved constructing an ML model to distinguish between non-clear cell RCCs and clear cell RCCs (ccRCCs) [21]. This model utilized radiomic features derived from the CT scans of 209 patients, leading to a classification system with accuracy rivaling expert radiologists. This achievement highlights the potential of radiogenomics to enhance cancer diagnostics and influence clinical decision-making [21]. These findings have profound implications. As radiogenomics evolves, it could lead to the discovery of biomarkers predicting treatment responses and long-term outcomes.

### 4.2. Summary of Critical Research Outcomes

Critical research in this field includes systematic reviews and meta-analyses leveraging MRI-derived radiomics features to grade clear cell RCC non-invasively. These studies demonstrate radiomics’ ability to significantly improve diagnostic accuracy in both localized and metastatic RCC. The aggregated data, with a median rRQS showing an upward trend, indicates strong odds ratios favoring radiomics in distinguishing benign tumors from RCC, underscoring its potential in refining renal tumor diagnostics and assessing treatment responses [19]. Additionally, advanced imaging techniques like CT-based volumetric radiomics have been crucial in guiding therapeutic decisions by predicting outcomes and treatment responses in RCC patients. The use of ML algorithms, particularly the XG Boost model, has allowed for the identification of associations with aggressive tumor characteristics, providing clinicians with crucial information on the potential behavior of large RCCs [22].

Adding to the body of evidence, Ferro et al. [23] discuss the current evidence and future prospects of radiogenomics in renal cancer management, highlighting the integration of radiomics features with genomics data. Their review underlines the potential of this approach while also acknowledging the limitations due to the retrospective design and small patient cohorts in clinical trials. Ferro et al. emphasize the necessity for well-designed prospective studies with larger patient populations to validate the promising results of radiogenomics and facilitate its transition into clinical practice [23].

In addition to these significant findings, Ferro et al. contribute to this evolving landscape with a comprehensive literature review that delineates AI’s and radiomics’ roles in enhancing the differentiation of benign and malignant kidney lesions, including subtypes of RCC, and in predicting genetic mutations and treatment responses, particularly in the context of metastatic RCC undergoing immunotherapy [24]. Their work underscores the expanding capabilities of neural networks in analyzing the vast and complex datasets derived from kidney imaging, offering quantitative insights into lesion contours, heterogeneity, and other critical features [24].

Radiogenomics research has also provided significant insights, linking imaging phenotypes with genetic profiles for a comprehensive approach to kidney-cancer management. This research has led to the identification of radiomic signatures associated with genetic alterations, informing targeted therapies tailored to individual tumor characteristics. 

In summary, AI and radiomics are set to significantly transform kidney-cancer care, heralding a future where treatments are personalized and more effective. As these technologies continue to advance and integrate into clinical practice, they hold great promise for revolutionizing kidney-cancer diagnosis, treatment, and prognosis.

## 5. AI and Radiomics in Prostate Cancer

### Development in Diagnostic and Prognostic Tools

The landscape of prostate-cancer diagnostics and prognosis has been significantly reshaped by the advancements in AI and radiomics. These technologies aim to provide more accurate detection, staging, and treatment planning. AI algorithms have particularly excelled in improving the detection of clinically significant PCa through MRI imaging, thereby reducing the dependence on invasive biopsies and enhancing tumor localization accuracy.

A study by Bleker et al. [25] aimed to evaluate the quality of multicenter studies on MRI radiomics for diagnosing clinically significant PCa. Following the PRISMA guidelines, it included multicenter studies and assessed their quality using the CLAIM and RQS checklists [25]. Only four studies were included, with an average total CLAIM score of 74.6% and an average RQS of 52.8%, leading to an average total quality score of 63.7% [25]. The study concluded that there have been very few multicenter radiomics PCa classification studies, mostly of average quality, highlighting the need for better-quality, preferably prospective studies and improved documentation for reproducibility and clinical utility. Sugano et al. in their review [26] aimed to assess the role of radiomics in (PCa) detection and evaluation and concluded that radiomics, involving high-throughput extraction of radiologic features from imaging, enhance clinical image utility in cancer management but despite potential advantages, consensus on radiomics implementation is lacking. With further validation and consensus-driven methodologies in larger, randomized studies, radiomics could significantly impact PCa management. 

Qiao et al. [27] investigated the application of biparametric magnetic resonance imaging (bpMRI) combined with machine learning (ML) techniques to predict the Ki67 index and Gleason grade group (GGG) in prostate cancer (PCa), with an emphasis on differentiating between indolent and invasive forms of the disease. The study included a cohort of 122 patients with histologically confirmed PCa. Radiomics features were extracted from T2-weighted imaging (T2WI), diffusion-weighted imaging (DWI), and apparent diffusion coefficient (ADC) maps. A variety of ML models, such as logistic regression (LR), support vector machine (SVM), random forest (RF), and K-nearest neighbor (KNN), were evaluated for their predictive performance [27]. Notably, the LR model utilizing ADC and T2WI was found to be the most accurate in predicting Ki67 expression, whereas the SVM model that integrated DWI and T2WI demonstrated the highest efficacy in forecasting GGG. These models have shown promise for the non-invasive identification of aggressive PCa, potentially aiding in the clinical decision-making process [27]. Figure 4 below by Qiao et al. [27] illustrates the process of feature extraction and model evaluation, highlighting the potential of bpMRI radiomics in the prognostic assessment of PCa.

Complementing this, radiomics has played a crucial role in extracting comprehensive information from imaging data, revealing subtle patterns and features indicative of tumor aggressiveness, stage, and potential therapy responses. These insights have been instrumental in developing predictive models for various clinical outcomes, including biochemical recurrence, disease progression, and responses to treatments like radiotherapy and hormone therapy [28,29]. In their study [28], Zhang et al. applied a bi-directional convolutional long short-term memory (CLSTM) network and radiomics analysis to DCE-MRI data for differentiating PCa from benign prostatic hyperplasia (BPH). Nine regions of interest (ROIs) were delineated using three methods to investigate the optimal peritumoral tissue amount for diagnosis [28]. The bi-directional convolutional long short-term memory (bi-directional CLSTM) with a ±20% region growing peritumoral ROI showed a higher mean AUC (0.89) compared to using the tumor without peritumoral tissue (AUC = 0.84) [28]. Deep learning consistently showed higher AUCs compared to radiomics, with a significant difference noted for the ±20% region growing peritumoral ROI (0.89 vs. 0.79 [28]. The study suggests that kinetic information from DCE-MRI, extracted via bi-directional CLSTM, could be valuable for PCa diagnosis [28]. 

Jaouen et al. [29] aimed to develop and test a zone-specific region-of-interest (ROI)-based computer-aided diagnosis system (CAD) for characterizing ISUP grade ≥2 prostate cancers on MRI. Using multi-vendor MRI datasets, the best models for peripheral and transition zones were selected and assessed in internal and external test datasets. The study compared the CAD system’s performance with the Prostate Imaging-Reporting and Data System version 2 (PI-RADSv2) scores. The CAD demonstrated similar accuracy to PI-RADSv2 in identifying ISUP ≥2 cancers, with comparable sensitivities and specificities in both internal and external test datasets [29]. The best models utilized the 25th ADC percentile in the transition zone and the 2nd ADC percentile and normalized the wash-in rate in the peripheral zone [29]. PI-RADSv2 AUCs were 82% and 86% in internal and external test datasets, respectively, similar to CAD AUCs [29]. CAD sensitivities were 86–89% and 90–91%, with specificities of 64–65% and 69–75% in internal and external datasets, demonstrating its effective performance in characterizing ISUP grade ≥2 prostate cancers [29].

The integration of AI with multiparametric MRI (mpMRI) is a hallmark of innovation in PCa diagnostics and strategy formulation for treatment. This synergy between AI algorithms and the detailed imaging data provided by mpMRI allows for an in-depth exploration of the tumor’s anatomy and functionality [30]. A comprehensive review of studies on radiomics in PCa, particularly focusing on the development and validation of radiomics models using MRI-derived image features, has provided a broad overview of the literature [30]. This review emphasizes models that hold high potential for clinical application, particularly those addressing crucial management concerns, while it also brings attention to the challenges of transitioning these models from research to clinical practice, underscoring the importance of ongoing collaboration between researchers and clinicians to refine these tools for enhancing patient care [30].

In the realm of prostate cancer diagnostics, Gentile et al. [31] have made significant strides with their study on the synergistic use of multiparametric magnetic resonance (mpMRI) and the Prostate Health Index (PHI). Their research, conducted on a cohort of 177 prostate cancer patients, leveraged artificial neural networks to develop a model that combines PHI and PI-RADS scores from mpMRI to distinguish between low- and high-Gleason-score cancers. The model, after being trained with 135 samples, demonstrated a sensitivity of 80% and a specificity of 68%, indicating the potential for a more precise initial diagnosis and personalized treatment approaches. The study suggests that further improvements could be achieved by training the model on larger datasets [31].

## 6. Challenges and Limitations

### 6.1. Discussing Current Challenges in Integrating AI and Radiomics

The integration of AI and radiomics into clinical cancer care, despite their transformative advancements, confronts several significant challenges. A primary hurdle is the standardization of imaging protocols and data. Variations in imaging equipment, settings, and techniques across different institutions can significantly impact the performance and generalizability of AI and radiomics models. This variability poses a substantial obstacle to achieving consistent and reliable results across various clinical settings [32,33]. Bi et al. in their article [32] discuss the role of AI in cancer care, particularly in medical imaging. They address how AI enhances the interpretation of cancer imaging, aiding in tumor delineation, predicting clinical outcomes, and assessing the impact of disease and treatment on adjacent organs [32]. The review also covers AI’s potential to automate initial image interpretations and influence clinical workflows in cancer detection and management across four cancer types (lung, brain, breast, and prostate) and highlights the need for the rigorous validation of AI tools in oncology for reproducibility and generalizability [32].

Data availability and quality are additional critical concerns. The development and refinement of AI algorithms heavily depend on high-quality, annotated datasets. However, the availability of such datasets is often limited, primarily due to privacy concerns, logistical challenges, and the costs associated with data collection and curation. This scarcity of quality data is a significant barrier to the advancement of AI algorithms [33]. Another challenge lies in integrating these technologies into existing healthcare workflows. Many clinical environments lack the necessary infrastructure or expertise to effectively deploy and utilize AI and radiomics tools. For clinicians to accurately interpret the outputs of these technologies and make informed decisions, a foundational understanding of AI and radiomics is essential [34]. Furthermore, AI and radiomics models often face the “black box” problem, where the decision-making process is not transparent or easily understandable. This lack of transparency can hinder the trust and acceptance of these tools among clinicians and patients, posing a significant challenge to their widespread adoption in clinical practice [35].

Table 1 provides a succinct comparison of the latest advancements in artificial intelligence (AI) and radiomics across three major urological cancers: bladder, kidney, and prostate cancer. It highlights key technological signs of progress, their impact on diagnosis and prognosis, and the challenges in integrating these innovations into clinical practice. This comparative analysis aims to encapsulate the transformative role of AI and radiomics in modern oncology, underscoring the unique applications and hurdles encountered in each cancer type.

### 6.2. Limitations of Existing Studies

While existing studies in AI and radiomics show promise, they are not without limitations. Many studies are retrospective and based on datasets from single institutions, which may not accurately represent the broader patient population. This raises concerns about the applicability and generalizability of their results [30,32]. The reproducibility of AI and radiomics studies is also a concern. The complexity of these models and the absence of standardized methodologies make it challenging to replicate results across different settings or datasets. This difficulty in reproducibility poses a significant barrier to the clinical translation of these technologies [33]. Addressing inherent biases in AI models is crucial. Biases can emerge from various sources, such as imbalanced datasets or biased algorithmic designs, leading to skewed or unfair outcomes. It is essential to identify and mitigate these biases to ensure the equitable and effective use of AI in oncology [34]. Lastly, the majority of current studies tend to focus more on the technical aspects of AI and radiomics, often neglecting the evaluation of clinical outcomes or patient-centric measures. This gap highlights the need for more research that directly assesses the impact of these technologies on patient care and outcomes, ensuring that technical advancements translate into tangible benefits for patients [35].

In conclusion, while AI and radiomics hold immense potential to transform cancer care, addressing these challenges and limitations is crucial for their successful and ethical integration into clinical practice. Ongoing research, collaboration, and innovation are essential to overcome these hurdles and fully leverage the benefits of these technologies in the field of oncology.

## 7. Conclusions

This review comprehensively highlights the significant advancements in the application of AI and radiomics in the diagnosis, prognosis, and treatment of bladder cancer, kidney cancer, and PCa. It underscores the transformation these technologies have brought to oncological diagnostics, prognostics, and therapeutic strategies, charting a new course in precision oncology. The insights gathered from this review are pivotal in understanding the impact of these technologies:For bladder cancer, AI and radiomics have precipitated a leap in diagnostic accuracy and staging acuity. The prowess of mpMRI and CT scans, fortified by these technologies, has been pivotal in discerning muscle invasion and pathological grades with unprecedented precision, laying the groundwork for tailored treatment regimens [34,36];Within kidney-cancer research, AI and radiomics have carved a niche in subtype differentiation and grade assessment, bolstered by their integration with radiogenomics. This synergy provides a multifaceted view of renal cell carcinoma, propelling forward the personalized treatment landscape [34];Prostate cancer management has been revolutionized by AI-augmented MRI, which has markedly improved tumor detection and localization. These advances, coupled with AI’s proficiency in prognosticating post-treatment trajectories, such as biochemical recurrence, are invaluable for patient-centric care.

### Implications for Future Research and Clinical Practice

The synergy between AI and radiomics is at the forefront of a transition towards more individualized cancer therapies. Through a deeper analysis of tumor biology, these tools can lead to the crafting of bespoke treatment plans poised to significantly improve patient prognosis. However, their adoption in everyday clinical practice introduces challenges, including the harmonization of imaging protocols, the assurance of data quality, and the need for transparent AI algorithms [34,36]. Looking ahead, research efforts should be channeled toward fostering multicenter collaborations that can extend the reach and validation of these technologies across diverse clinical settings. In parallel, there is a pressing requirement to develop AI platforms that clinicians find intuitive and that integrate effortlessly with existing healthcare systems. Tackling ethical issues, particularly those relating to data security and the prevention of bias in AI algorithms, is crucial for the ethical application of these technologies [20]. Additionally, the field must pivot towards patient-centered research that measures the tangible effects of AI and radiomics on patient well-being and survival. Conducting such research is vital to affirm the clinical relevance of these innovations and to guarantee they address the genuine needs of those they are designed to serve.

## Figures and Tables

**Figure 1 cancers-16-00810-f001:**
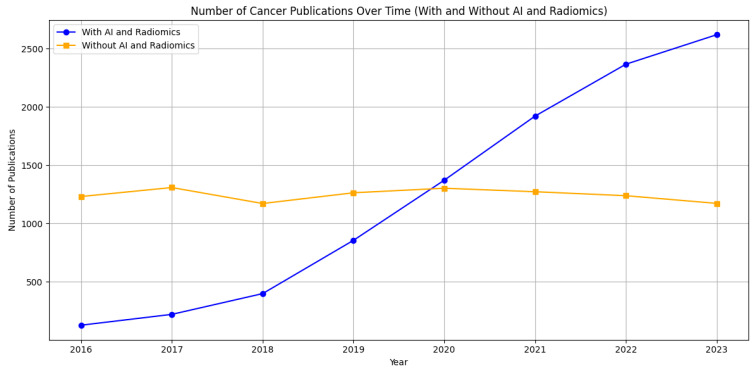
Comparative trends in cancer research publications from 2016 to 2023 on PubMed. The graph shows a marked increase in publications incorporating artificial intelligence and radiomics (blue line), contrasting with a relatively steady count of publications in broader cancer research (orange line). This illustrates the burgeoning role of advanced computational methods in cancer-related studies.

**Figure 2 cancers-16-00810-f002:**
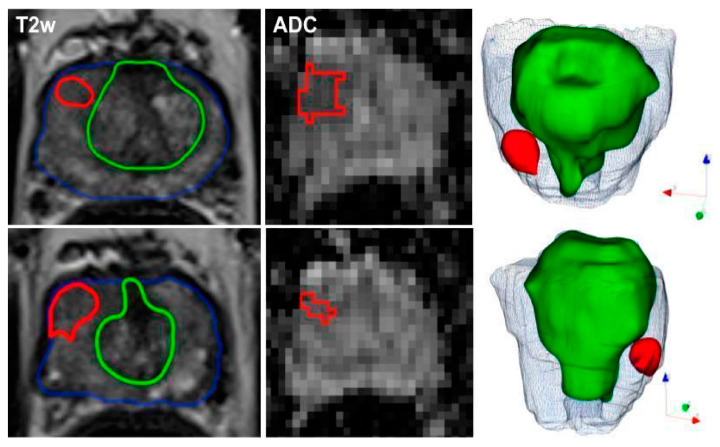
Representative segmentation of a patient on axial T2-weighted and ADC-images with an initial prostate-specific antigen level of 12.9 ng/ml, normal DRE result, and a highly suspicious lesion located medio-apically in the right peripheral zone (PI-RADS category 4). Targeted MRI/ultrasound-fusion biopsy confirmed the presence of PCa, Gleason 3+4, ISUP 2. The contours outline the following manual segmentations: blue—whole organ, green—transition zone, red—index lesion.

**Figure 3 cancers-16-00810-f003:**
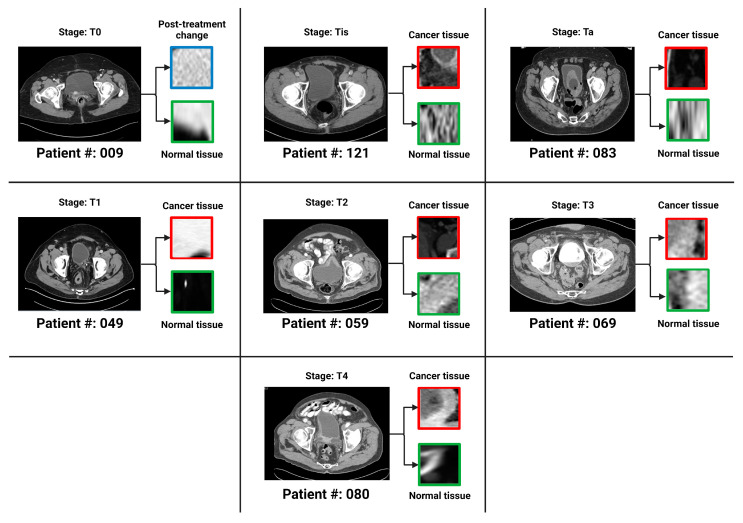
One bladder CT scan per stage (along with the corresponding regions of interest that were used in the various classification tasks) has been provided. The seven stages of urothelial carcinoma analyzed in the study are *Ta*, *Tis*, *T*0, *T*1, *T*2, *T*3 and *T*4 (*T*0 has not been shown in the figure because *T*0 represents a stage where the tissue of interest shows no evidence of malignancy).

**Figure 4 cancers-16-00810-f004:**
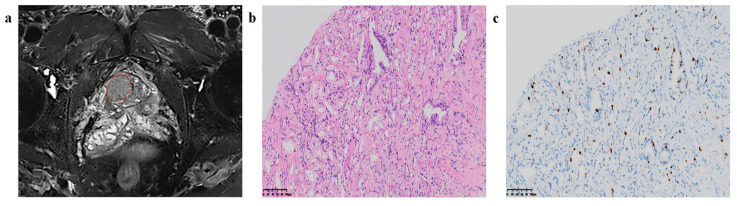
Region of interest (ROI) segmentation and corresponding pathological pictures. (**a**) The red circle is marking the ROI segmentation, (**b**) Loupe image of hematoxylin-eosin stain shows the GS = 4 + 3 (×20), (**c**) Loupe image of immunohistochemical stain shows the percentage.

**Table 1 cancers-16-00810-t001:** Overview of AI and radiomics advances in urological cancers: a comparative analysis of bladder, kidney, and prostate cancer.

Cancer Type	Key Advances in AI/Radiomics	Diagnostic/Prognostic Impact	Challenges in Integration
Bladder Cancer	- Utilization of AI and radiomics for enhanced imaging and predictive modeling.	- Improved diagnosis and staging, particularly in predicting muscle invasiveness and pathological grades.	- Need for standardization and overcoming the “black box” nature of AI.
Kidney Cancer	- Application of AI in differentiating RCC subtypes and grades. - Integration of radiogenomics for comprehensive disease profiling.	- Better tumor characterization leading to more tailored treatments.	- Challenges include data availability and integrating these technologies into existing healthcare systems.
Prostate Cancer	- AI-enhanced MRI for improved tumor detection and localization. - Development of predictive models for treatment outcomes.	- Reduced reliance on invasive biopsies, and more effective treatment planning.	- Issues with the interpretability and transparency of AI models, necessitating multicentric collaborations for broader validation.

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
