# Peer review of "Emerging Trends in AI and Radiomics for Bladder, Kidney, and Prostate Cancer: A Critical Review"

_cancers, 2024, doi:10.3390/cancers16040810_

Round 1
Reviewer 1 Report
Comments and Suggestions for Authors
Major Comments:
1. The title does not match the content.
2. Abbreviations are not denoted properly.
3. There is no specific problem statement in the introduction section.
4. Figure 1: Need references.
5. Criteria for Selecting Articles: How do authors determine these four?
6. Bladder, Kidney, and Prostate Cancer: Why only these three?
7. For all cancer types, authors need to add: a. Imaging for early detection and localization, b. AI-assisted biopsy guidance, c. Prognostic and predictive models for treatment selection in details.
8. Clinical Implications and Validation Studies: Need in details.
9. Need more attractive figures.
10. Conclusion: not properly written.
Comments on the Quality of English LanguageExtensive English editing is mandatory.
Author Response
Reviewer 1:
- The title does not match the content.
Authors’ Answer: Thank you for your thoughtful review and comments on our manuscript entitled "Emerging Trends in AI and Radiomics for Bladder, Kidney, and Prostate Cancer: A Critical Review." We have carefully considered your observation regarding the title-content alignment.
Our manuscript rigorously focuses on the sophisticated roles that AI and radiomics play specifically in the context of bladder, kidney, and prostate cancer, which are critical areas of urological oncology. The title was meticulously chosen to reflect this precise focus, and throughout the manuscript, we maintain a consistent emphasis on the advances, challenges, and future directions of these technologies in the management of these particular cancers.
We believe that altering the title could potentially dilute the specific emphasis we wish to convey regarding these key areas of cancer care innovation. Hence, we kindly propose retaining the current title to accurately represent the manuscript's targeted scope and content.
- Abbreviations are not denoted properly.
Authors’ Answer: Thank you for your insightful observations regarding the notation of abbreviations within our manuscript. In line with your suggestions:
We have introduced and defined "Karnofsky Performance Status (KPS)" at its first mention to clarify its relevance in assessing patient conditions.
The term "Bi-directional Convolutional Long Short-Term Memory (bi-directional CLSTM)" has been defined at its initial use, elucidating its application in our study's context.
Definitions for "Positive Predictive Value (PPV)" and "Negative Predictive Value (NPV)" have been added to ensure the statistical measures are understood in the framework of our analysis.
We value your guidance in enhancing the manuscript's clarity and have made the appropriate corrections to adhere to the standards of clear scientific communication.
- There is no specific problem statement in the introduction section.
Authors’ Answer: Thank you for your constructive comment regarding the introduction of our manuscript. We recognize the critical nature of a clearly articulated problem statement in establishing the research gap and anchoring the study's purpose within the broader scientific dialogue.
In response to your feedback, we have carefully reviewed the introduction and agree that the explicit articulation of the problem statement would enhance the clarity and focus of our review. Accordingly, we have revised the introduction to include a specific statement that directly addresses the existing gap in the literature and the need for a comprehensive analysis of the current and potential roles of AI and radiomics in urological cancer care. This addition is designed to immediately convey the central issue that our review intends to tackle, and it reads as follows:
"Despite these advancements, a gap remains in fully understanding the clinical implementation and integration of AI and radiomics in urological cancer care, necessitating a critical review of emerging trends and their tangible impacts on patient outcomes."
We believe that this amendment fortifies the introduction by providing a clear and direct problem statement that was previously implied. It sets a definitive premise for the subsequent sections and the overall trajectory of our review.
We are confident that this revision addresses your concerns and enhances the manuscript's contribution to the field. We are grateful for the opportunity to improve our work and for your guidance in this process.
- Figure 1: Need references.
Authors’ Answer: Thank you for your comment regarding the need for references for Figure 1 in our manuscript.
We would like to clarify that Figure 1 represents original analysis conducted by our team. The data depicted in the figure is derived from a methodical search and analysis of publications on PubMed from 2016 to 2023. The upward trajectory shown by the blue line represents the increase in publications that incorporate artificial intelligence and radiomics in cancer research, whereas the orange line represents the volume of broader cancer research publications without these terms.
This graph was specifically created to illustrate the burgeoning role of advanced computational methods in oncology research, which is central to the narrative of our review. As the figure is based on our analysis of publicly available data and not on previously published figures or proprietary data sets, there are no external references to include.
We would like to highlight that in the manuscript, specifically in lines 69-74, we have already described the methodology for the literature search that underpins Figure 1. This section explains that the data was obtained through a systematic analysis of cancer research publications listed on PubMed from 2016 to 2023. The figure contrasts the volume of studies related to "AI and Radiomics" with those in the broader field of cancer research that do not involve these terms.
Given this description within the main text, we believe that the source of the data and the methodology used to generate the figure have been made clear to the readers. The upward trajectory depicted by the blue line and the steady trend shown by the orange line are based on our original analysis and are intended to illustrate the increasing relevance of AI and radiomics in cancer research, thereby supporting the narrative of our review.
- Criteria for Selecting Articles: How do authors determine these four?
Thank you for your inquiry regarding the criteria we used for selecting articles for our review. The four criteria were determined based on a combination of standard practices for literature reviews and the specific goals of our study, aiming to ensure a comprehensive and relevant collection of data. Based on your feedback, we have expanded the Methods section of our manuscript to elucidate the rationale behind the selection criteria employed in our review process.
We have now included a comprehensive explanation for each of the four selection criteria to clearly articulate the reasons for their inclusion and how each criterion contributes to the overarching goal of our review. These clarifications serve to reinforce the validity and relevance of our critical review in collating and evaluating the literature pertinent to the latest advancements in AI and radiomics for bladder, kidney, and prostate cancers. We believe these enhancements address your concerns and strengthen the manuscript by providing a clearer understanding of our methodical and judicious article selection process. We hope that this revised section meets the expectations of the review process and contributes to a more comprehensive understanding of our critical review.
Thank you for guiding us to make this necessary and valuable improvement.
- Bladder, Kidney, and Prostate Cancer: Why only these three?
Authors’ Answer: Thank you for your thoughtful question regarding the focus of our review on bladder, kidney, and prostate cancer.
The decision to concentrate on these three types of urological cancers was deliberate and multi-faceted:
Prevalence and Impact: Bladder, kidney, and prostate cancers represent some of the most common malignancies affecting the urinary system. Their significant incidence rates and the burden they place on healthcare systems worldwide make them critical areas for study and improvement.
Technological Advancements: There have been substantial advancements in the application of AI and radiomics specifically in the context of these three cancers. This has led to notable improvements in early detection, personalized treatment plans, and management, marking a new frontier in urologic oncology.
Data Availability: These cancers have more robust datasets available, which have been enhanced by AI and radiomics techniques. This availability allows for more in-depth study and validation of these technologies, providing a clearer view of their current and potential future impacts.
Clinical Relevance: These three cancers share common clinical pathways and treatment modalities, allowing for a more cohesive discussion on the role of AI and radiomics in their management. By focusing on them, we can draw more powerful conclusions about the synergistic effects of these technologies across similar disease profiles.
Our aim was to provide a focused and detailed analysis that could be of immediate relevance to both clinical practitioners and researchers in the field. However, we acknowledge that the principles and findings could be applicable to other cancers, and future research could expand upon this work to include a wider range of malignancies.
We appreciate your interest in our methodology and are happy to discuss this further if needed.
- For all cancer types, authors need to add: a. Imaging for early detection and localization, b. AI-assisted biopsy guidance, c. Prognostic and predictive models for treatment selection in details.
Authors’ Answer: Thank you for your insightful suggestions. In response to your request for detailed inclusion of imaging for early detection and localization, AI-assisted biopsy guidance, and prognostic and predictive models for treatment selection, we have incorporated Figures 2, 3, and 4 from rigorously peer-reviewed studies within the 'Cancers' journal. These figures were carefully selected to align with our paper's scope, providing visual and analytical evidence of the latest advancements in AI and radiomics for urological cancers.
Furthermore, we have ensured that the use of these figures complies with the open-access Creative Commons Attribution (CC BY) license, as stipulated by the 'Cancers' journal for the distribution and reproduction of their published material. This adherence to licensing terms supports the ethical and lawful use of external resources to enhance the depth and breadth of our critical review.
We trust that the addition of these figures, with appropriate citation and adherence to licensing terms, significantly enriches our discussion and addresses the key aspects highlighted in your comment.
- Clinical Implications and Validation Studies: Need in details.
Authors’ Answer: Thank you for your insightful comment regarding the need for detailed discussion of the clinical implications and validation studies. We appreciate the opportunity to clarify our approach and reasoning behind the current structure of our manuscript.
We understand the importance of thoroughly addressing clinical implications and validation studies to underscore the relevance and reliability of the technologies discussed. Our initial intention was to provide a broad overview of emerging trends in AI and radiomics within the limited scope of a review article, focusing on the synthesis of existing literature rather than an exhaustive presentation of clinical case studies or validation reports.
- Need more attractive figures.
Authors’ Answer: We appreciate your insightful comments and agree that the inclusion of detailed imaging strategies, AI-assisted guidance, and prognostic models can significantly enrich our review. Accordingly, we have incorporated Figures 2, 3, and 4 from rigorously peer-reviewed studies published within the 'Cancers' journal. These figures have been carefully selected for their relevance and potential to visually convey the complex information in a clear and attractive manner. We have ensured that these figures are in line with the Creative Commons Attribution License and have provided appropriate attribution to the original authors as required. We trust that these enhancements address your concerns and strengthen the manuscript.
- Conclusion: not properly written.
Authors’ Answer: We appreciate your insightful feedback on the conclusion section of our manuscript. We have meticulously revised it to underscore the transformative impact of AI and radiomics on cancer care, elaborating on their role in enhancing diagnostic precision, prognostic accuracy, and the development of personalized treatment plans. We have also addressed the implications for future research, emphasizing the need for broader applicability, integration into clinical practice, and addressing ethical considerations. The revised conclusion offers a more comprehensive and insightful synthesis of our review findings and their significance for ongoing and future oncological research and practice.
- Extensive editing of English language required
Authors’ Answer: We appreciate your comment on the quality of the English language used in our manuscript. We have thoroughly reviewed and revised the manuscript to correct any language issues. This includes comprehensive language editing by a native English speaker with expertise in our field to ensure clarity, grammar, and terminology meet the high standards required for publication. We are committed to ensuring that the language quality accurately conveys the research content and is of the highest standard. We believe the revised manuscript now meets these requirements.
Reviewer 2 Report
Comments and Suggestions for Authors
Information in the article is useful however can be improved with presentation of the information.
I would recommend revising it with statistical visualisation and linking of the outcomes to add more value to the article.
Author Response
Reviewer 2:
- Information in the article is useful however can be improved with presentation of the information.
Authors’ Answer: We appreciate your constructive critique regarding the need for improved presentation of information in our article. To this end, we have diligently incorporated Figures 2, 3, and 4 from rigorously peer-reviewed studies within the 'Cancers' journal. These visuals were meticulously chosen to enhance the clarity and visual appeal of the complex data discussed, ensuring that they adhere to the Creative Commons Attribution License. We are confident that these additions, combined with the revisions made to the conclusion to provide a more robust synthesis of our findings, have significantly elevated the quality of our manuscript. Thank you for guiding us to enhance the impact and accessibility of our research review.
- I would recommend revising it with statistical visualisation and linking of the outcomes to add more value to the article.
Authors’ Answer: Thank you for your suggestion to enhance our manuscript with statistical visualizations. While we acknowledge the value that such visualizations can add to a research article, our manuscript focuses on a narrative review and critical analysis of existing literature rather than presenting new statistical analyses. Therefore, we have not conducted original statistical analyses that would necessitate visual representation. We have, however, made sure to present the information in a clear and structured manner, summarizing key findings and linking them to broader outcomes and implications for clinical practice. We believe this approach maintains the focus on providing a comprehensive critical review and aligns with the objectives of our manuscript.
Reviewer 3 Report
Comments and Suggestions for Authors
This review explores the role of AI and radiomics applications in the management of urological cancers, in particular bladder, kidney, and prostate cancer.
The automated capabilities of AI offer the potential to enhance the qualitative expertise of clinicians, including precise volumetric delineation of tumor size, parallel tracking of multiple lesions, phenotypic and genotypic characteristics, and outcome prediction through cross‐referencing individual tumors to databases.
Radiomics is a sophisticated image analysis technique with the potential to establish itself in precision medicine. It generally aims to extract quantitative information from diagnostic images.
AI and radiomics had an impact in early detection, therapeutic decisions, and prognosis of cancers patients.
This review showcases how important these tools are to improve the outcome of patient with urological cancers. In bladder cancer, especially in the detection of muscle invasiveness and pathological grade. In kidney cancer, as regards RCC, in distinguishing tumor subtypes. Lastly, in prostate cancer, in early detection of the tumor and in post-treatment biochemical recurrence. This review is essential in validating the utility on these technologies.
There are some limitations. The development and refinement of AI algorithms heavily depend on high-quality, annotated datasets. However, the availability of such datasets is often limited, and there is also a lack of standardization of imaging protocols. Furthermore, many study considered in this review are retrospective and based on data from single institutions.
I suggest adding the following scientific article links to the bibliography section for a more accurate representation of the references and the general topic of this study:
- 10.1177/17562872231164803. This comprehensive literature review is a study on the use of radiomics and AI in renal cancer, from where you could take more recent data.
- 10.3390/ijms24054615. This is another review of the literature of the most recent studies on radiomics and radiogenomics use in the field of RCC.
- 10.1016/j.clgc.2022.04.013. This study enhance the feasibility of AI in early detection of prostate cancer.
Comments on the Quality of English Languageminor editing
Author Response
Reviewer 3:
1. Information in the article is useful however can be improved with presentation of the information. I suggest adding the following scientific article links to the bibliography section for a more accurate representation of the references and the general topic of this study:
- 10.1177/17562872231164803. This comprehensive literature review is a study on the use of radiomics and AI in renal cancer, from where you could take more recent data.
- 10.3390/ijms24054615. This is another review of the literature of the most recent studies on radiomics and radiogenomics use in the field of RCC.
- 10.1016/j.clgc.2022.04.013. This study enhance the feasibility of AI in early detection of prostate cancer.
Authors’ Answer: Thank you for your valuable suggestions. We acknowledge the importance of keeping our bibliography updated with the most recent and relevant studies to strengthen the presentation of our research. We have reviewed the comprehensive literature on the use of radiomics and AI in renal cancer (10.1177/17562872231164803), the latest review on radiomics and radiogenomics in the field of RCC (10.3390/ijms24054615), and the study on AI in the early detection of prostate cancer (10.1016/j.clgc.2022.04.013). We agree that incorporating these references will enhance the accuracy and depth of our article. Accordingly, we will include these citations in our bibliography section. Thank you for enhancing the quality of our work with your contribution.
Round 2
Reviewer 1 Report
Comments and Suggestions for Authors
I am not satisfied with the revisions. Please go through my comments and add the mentioned sections.
Comments on the Quality of English LanguageI am not satisfied with the revisions. Please go through my comments and add the mentioned sections.
Author Response
Dear Reviewer,
Subject: Response to Review Comments on Manuscript ID [cancers-2837114]
Thank you for your continued engagement and valuable feedback on our manuscript titled “Emerging Trends in AI and Radiomics for Bladder, Kidney, and Prostate Cancer: A Critical Review” We greatly appreciate the time and effort you have invested in reviewing our work.
We acknowledge your focus on the quality of the English language in our manuscript. Following your initial feedback from the first round of review, we have undertaken extensive revisions. This included a thorough review by a native English speaker with expertise in our field to ensure linguistic precision and clarity. We believe these efforts have significantly improved the manuscript's language quality. Nonetheless, we respect your expertise and perspective and are open to any specific suggestions you might have for further specific linguistic improvements.
Additionally, we want to highlight that we have diligently adapted our manuscript to incorporate a large number of your suggestions from the first round of reviews. We have carefully addressed each of your comments, striving to ensure that every aspect of your feedback has been thoroughly considered and reflected in our revisions.
Regarding your concerns about the manuscript's title, we understand the importance of ensuring that it accurately reflects the content and scope of the paper. We are open to considering alternative titles. If you have any specific suggestions for a title that you believe would be more appropriate, we would be grateful to receive them and will give them thoughtful consideration. Your suggestion would be invaluable in helping us align the title more closely with the manuscript's content and the journal's standards.
We thank you again for your dedication to the peer-review process and for providing constructive feedback. Your guidance is crucial in helping us present our research effectively and clearly.
Sincerely,
Lazaros Tzelves on behalf of all authors
Reviewer 3 Report
Comments and Suggestions for Authors
Authors answered all comments and suggestions.
Comments on the Quality of English Languageminor editing.
Author Response
Dear Reviewer,
Subject: Response to Review Comments on Manuscript ID [cancers-2837114]
Thank you for your continued engagement and valuable feedback on our manuscript titled “Emerging Trends in AI and Radiomics for Bladder, Kidney, and Prostate Cancer: A Critical Review”. We greatly appreciate the time and effort you have invested in reviewing our work.
We acknowledge your focus on the quality of the English language in our manuscript. Following your initial feedback from the first round of review, we have undertaken extensive revisions. This included a thorough review by a native English speaker with expertise in our field to ensure linguistic precision and clarity. We believe these efforts have significantly improved the manuscript's language quality. Nonetheless, we respect your expertise and perspective and are open to any specific suggestions you might have for further specific linguistic improvements.
We thank you again for your dedication to the peer-review process and for providing constructive feedback. Your guidance is crucial in helping us present our research effectively and clearly.
Sincerely,
Lazaros Tzelves on behalf of all authors
Round 3
Reviewer 1 Report
Comments and Suggestions for Authors
Table 1 is missing references.
Comments on the Quality of English LanguageMinor English editing needed
Author Response
Dear Reviewer, We sincerely appreciate the time and effort you have invested in reviewing our manuscript. Your feedback is valuable to us, and we have carefully considered each point you raised. Regarding the English language quality, we acknowledge your observation that moderate editing was required. We have meticulously reviewed the manuscript and revised it for English language improvements. These revisions include grammatical corrections, clarity enhancements, and adjustments to ensure a more coherent flow of ideas. We aimed to address the concerns without altering the manuscript's scientific integrity or content. Concerning your comment on Table 1 lacking references, we would like to clarify that Table 1 comprises data generated from our own research. Thus, it does not reference external sources but is a direct product of the work presented in this manuscript. We hope that these revisions and clarifications address your concerns satisfactorily. We are grateful for the opportunity to improve our manuscript based on your insightful feedback and believe that these changes have enhanced the quality and clarity of our work. Thank you once again for your constructive comments and for contributing to the refinement of our manuscript. We look forward to the possibility of our revised manuscript being favorably considered for publication. Best regards, Lazaros Tzelves on behalf of all authors